# Trajectory Optimization with Complex Obstacle Avoidance Constraints via Homotopy Network Sequential Convex Programming

Wenbo Li [1], Wentao Li [2], Lin Cheng [2] and Shengping Gong [2,*]

[1] School of Aerospace Engineering, Tsinghua University, Beijing 100084, China
[2] School of Astronautics, Beihang University, Beijing 100191, China
* Correspondence: gongsp@buaa.edu.cn

**Abstract:** Space vehicles' real-time trajectory optimization is the key to future automatic guidance. Still, the current sequential convex programming (SCP) method suffers from a low convergence rate and poor real-time performance when dealing with complex obstacle avoidance constraints (OACs). Given the above challenges, this work combines homotopy and neural network techniques with SCP to propose an innovative algorithm. Firstly, a neural network was used to fit the minimum signed distance field at obstacles' different "growth" states to represent the OACs. Then, the network was embedded with the SCP framework, thus smoothly transforming the OACs from simple to complex. Numerical simulations showed that the proposed algorithm can efficiently deal with trajectory optimization under complex OACs such as a "maze", and the algorithm has a high convergence rate and flexible extensibility.

**Keywords:** real-time trajectory planning; sequential convex programming; homotopy technique; deep neural network

## 1. Introduction

With the increasing complexity of aerospace missions, the autonomous online operation of the vehicle has become an essential part of advanced missions. Automatic guidance methods aim to generate guidance commands in real-time to meet various constraints [1]. Automatic guidance methods represented by real-time trajectory optimization (RTO) have attracted many scholars. For example, in the rocket recovery guidance problem, the RTO method can effectively increase the feasible region of the initial states and realize a safe landing [2]. In addition, due to the unknown environment of a wild quadrotor swarm scenario, it is vital to obtain the control variables in real-time to avoid collisions between quadrotors and obstacles [3]. RTO has become one of the most promising options for this type of task.

In recent years, the sequential convex programming (SCP) algorithm has progressed significantly as one of the RTO methods [4–6]. Liu first proposed SCP in [7,8]. It converts nonconvex constraints into affine constraints by successive linearization, thus constructing a series of second-order cone programming (SOCP) subproblems. The local optimal solutions satisfying the primal nonconvex constraints can be obtained by successively solving the subproblems. Since one can solve each subproblem in polynomial time using the mature interior point method solver, SCP becomes an efficient method. On this basis, Mao [9] and Bonalli [10] proposed the SCP algorithms Scvx and GuSTO, enjoying certain convergence guarantees. Unfortunately, it has been suggested that when the convergent virtual control is non-zero, the obtained solution is infeasible for the original problem [4,11].

In practical scenarios, spacecrafts and quadrotors will encounter a variety of complex obstacle avoidance constraints (OACs) [12]. Augugliaro [13] and Morgan [14] simplified the no-fly zone to a sphere, and then converted the OACs into a series of affine constraints

by linearization. However, this method is limited to obstacles of simple shapes. Virgili-Llop [15] proposed the concept of the minimum signed distance (MSD) field to deal with obstacles of arbitrary shapes. Zhang [16] and Misra [17] decomposed the OACs into the sum of a convex polynomial function and a concave polynomial function via the convex–concave procedure algorithm, thus preserving the higher-order information of OACs. Richards [18,19] modeled the problem as mixed-integer linear programming (MILP) and introduced binary variables to deal with OACs. However, the MILP runtime varies exponentially with the number of binary variables. Space vehicles' hardware often finds it hard to support such large-scale computations [11]. Szmuk [20] introduced the continuous state-triggered constraint (CSTC), applied to the obstacle avoidance flight in the powered landing phase of a 6-DOF rocket. Then, Szmuk extended the CSTC method to a compound CSTC and applied it to quadrotor trajectory planning [21]. Unfortunately, the above SCP-based algorithms still have many shortcomings. For example, the convergent solution cannot be guaranteed to be feasible. It is even tricky for traditional SCP to obtain feasible solutions under complex obstacles [22]. Besides, the selection of reference trajectories significantly impacts the optimality and feasibility of the solutions [1].

In recent years, with the development of homotopy and neural network techniques, many new methods and ideas have emerged in the field, providing new perspectives to overcome such shortcomings:

1. The general idea of the homotopy technique is to solve a series of simple transition problems so that the solution of the transition problem gradually approaches the optimal solution of the primal problem. This method reduces the problem's difficulty, thereby significantly improving the success rate of trajectory planning [4,23–27]. Taheri [23,24] and Saranathan [25] first used the homotopy method to convert a multi-point boundary value problem into an easier-to-solve two-point boundary value problem, thereby dealing with multiple propulsion modes and dynamic environments. Malyuta [26] combined the homotopy method with the SCP algorithm to solve trajectory planning in a rendezvous maneuver under the constraints of discrete logic.
2. As an auxiliary tool for trajectory optimization, the neural network can effectively improve the performance of traditional algorithms [28–33]. Yin [28] proposed a trajectory planning method based on a neural network, thus improving the indirect method's convergence rate. Tang [29] and Banerjee [31] used the neural network to fit the initial reference trajectory offline. The SCP algorithm iterates from the reference trajectory predicted by the network, which can effectively reduce the number of iterations. Li [30] used a neural network predictor to estimate the optimal flight time. Only one SOCP needs to be solved online, significantly improving real-time performance.

Unlike the above methods, we combined the homotopy and neural network techniques synchronously and proposed an RTO algorithm, called the homotopy network sequential convex programming algorithm (HNSCP), that can deal with general complex OACs. First, a universal obstacle "growth" method, namely the obstacle interface regression (OIR) approach, was proposed. Then, a homotopy neural network (HNN) was subsequently proposed. The inputs of the HNN are the spatial position coordinates and the homotopy parameter, and the output is the MSD. After that, Bayesian regularization was used to train the HNN. Finally, the OACs represented by the HNN were embedded into the SCP algorithm by successive linearization.

It is worth noting that, in past work, the STOMP [34] and CHOMP [35] algorithms adopted the strategy of computing the signed distance field offline and then applying it online without introducing a neural network. This strategy can theoretically be extended and applied to this study, but there are some difficulties in practice.

The STOMP and CHOMP algorithms only compute the MSD field of a single map. However, after introducing the homotopy technique, it needs to compute the MSD fields under a series of homotopy parameters, which will greatly increase the storage burden. In Example 2, the homotopy parameter increment selected was 0.02. This indicates that the MSD field information of 51 maps will need to be stored successively, and the storage

burden will increase dramatically. Therefore, it is advantageous to introduce the neural network to fit the MSD field.

We still take Example 2 to illustrate. Assuming that the strategy of STOMP and CHOMP is adopted to build a sign distance field with an accuracy of 2.5 m (rather rough) offline, then the method needs to store $8 \times 10^6$ parameters. On the contrary, the proposed method only needs to store 3400 parameters. The amount of data stored is reduced to 0.0425%. Although the current algorithm only considers the three-dimensional position space constraints, the algorithm can be extended to the six-dimensional state space constraints without much effort to address more general constraints. The traditional storage strategy will lead to a sharp increase in the storage burden, and the advantage of the HNN will be more significant.

Our work has two major innovations:

1. A universal obstacle regression method, namely OIR, is proposed. To the best of our knowledge, this is the first attempt to apply OIR to the trajectory planning field.
2. The concept of the HNN is proposed for the first time. Compared with [28–33], training the HNN does not require solving optimal control problems. Only the MSD field is needed so that adequate samples can be generated efficiently and stably.

Our algorithm also has the following three major advantages:

1. After introducing OIR and the HNN, the algorithm supported the modeling of arbitrary complex OACs, thus greatly expanding the application scenarios of SCP.
2. The homotopy parameter can be flexibly updated by changing the input parameters of the HNN, thereby realizing the smooth transition from simple obstacles to complex ones. The convergence and optimality of the SCP algorithm were greatly improved. Numerical simulations showed that the HNSCP algorithm converges well, even with the most straightforward linear interpolated initial reference trajectory.
3. After introducing the HNN, MSD fields with different homotopy parameters could be fitted with only a small amount of data. The data storage was significantly reduced, which is vital for online applications.

The structure of this paper is as follows. Section 2 presents a mathematical description of the optimal control problem. Section 3 proposes the HNSCP algorithm, which can be divided into two procedures: online and offline. Section 4 describes the details of the online procedure, including convexification and the successive iteration algorithm. Section 5 introduces the details of the offline procedure, including the OIR and HNN training. In Section 6, a simulation of two specific examples is conducted. The first example compared the proposed algorithm with state-of-the-art algorithms, verifying that the proposed algorithm had a better performance. The second example increased the complexity of the obstacle. A complex "maze" was designed to demonstrate the algorithm's adaptability in complex environments. Section 7 presents the main conclusions.

## 2. Problem Statement

This work focuses on dealing with complex constraints, and therefore does not establish an accurate dynamic model of the vehicle and does not limit the type of the vehicle. Readers may consider the "vehicle" as a rocket, quadrotor, or planetary lander in the following passage. First, a reference coordinate system was established, as shown in Figure 1. The origin of this coordinate system is stationary relative to the ground. The *y*-axis points to the zenith, and the *x*-axis and *z*-axis are parallel to the ground and form a right-handed rectangular coordinate system, where the *x*-axis is in the drawing plane. The *z*-axis is perpendicular to the drawing plane.

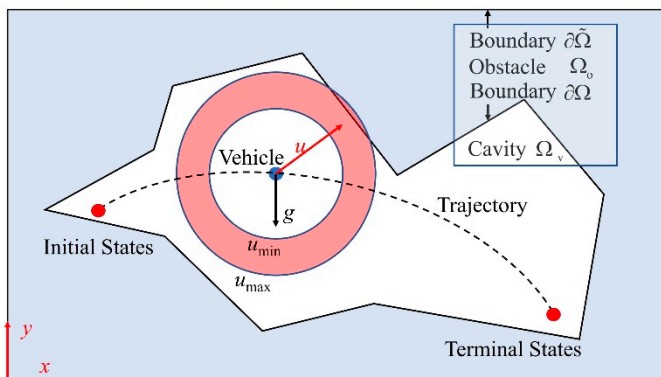

**Figure 1.** Definition of the coordinate system for trajectory-planning problem.

Secondly, the dynamic equations are listed. The vehicle was regarded as a mass point and only the thrust and gravity were considered. The 3-DOF dynamic equations are as follows:

$$\begin{cases} \dot{p} = v \\ \dot{v} = u + g \end{cases} \tag{1}$$

where $p : R_{++} \to R^3$ represents the position vector of the vehicle, $v : R_{++} \to R^3$ represents the velocity vector, and $u : R_{++} \to R^3$ represents the thrust acceleration vector, which was also the control vector. $g \in R^3$ is the gravity acceleration vector and satisfies $g = [0, -g, 0]^T$, where $g = 9.8065$ m/s$^2$. Although this work only considered the thrust and gravity, the proposed method can be extended and is applicable to more accurate dynamic models or 6-DOF models. This is beyond the scope of the study and will be left to future publications.

Then, two kinds of process constraints were imposed.

(1) Due to the limitations of the vehicle's own propulsion system, it was necessary to impose limits on the magnitude of the thrust acceleration:

$$u_{min} \leq \|u\| \leq u_{max} \tag{2}$$

where $\|u\|$ is the Euclidean norm of vector $u$, which represents the thrust acceleration magnitude. In the following text, $\|u\|$ will be abbreviated as $u$. $u_{min}$ and $u_{max}$ are the lower and upper bounds of the thrust acceleration, respectively, as shown in Figure 1.

(2) OACs were considered. For the convenience of description, we divided the research region into two parts. The impenetrable obstacle was denoted by $\Omega_o$, as shown in the blue part of Figure 1. The traversable cavity was represented by $\Omega_v$, as shown in the white part of Figure 1. The boundary between the obstacle and the cavity was denoted by $\partial\Omega$. The boundary between the research region and the outside region was denoted by $\partial\tilde{\Omega}$. Then, the OACs can be written as:

$$p \in \Omega_v \tag{3}$$

The necessary initial state constraints and terminal state constraints were imposed:

$$\begin{cases} p(t_0) = p_0 \\ p(t_f) = p_f \\ v(t_0) = v_0 \\ v(t_f) = v_f \end{cases} \tag{4}$$

where $t_0$ and $t_f$ represent the initial and terminal times, respectively. $p_0$ and $v_0$ are the position and velocity vectors at the initial time, respectively. $p_f$ and $v_f$ are the position and velocity vectors at the terminal time, respectively. $t_f$ is treated as an unknown variable, indicating that the optimal control problem with free terminal time is solved.

Finally, the velocity increment was selected as the cost function. Minimizing this cost function can save the fuel consumption of a rocket or the energy consumption of a quadrotor:

$$J = \int_{t_0}^{t_f} u \, dt \tag{5}$$

The mathematical description of the optimal control problem is shown in Problem 1.

Problem 1:

Cost function: $\underset{t_f, \boldsymbol{u}(t)}{\text{minimize}}$ (5)

Subject to: (1)~(4)

## 3. HNSCP Algorithm

Traditional sequential convex programming (TSCP) is widely used in [13–15]. Unfortunately, the TSCP algorithm can only deal with obstacles with simple geometric shapes. Reference [22] shows that when the shape of an obstacle is complex, sub-SOCP may be infeasible, and the iteration will be interrupted. Eventually, the convergence rate of the algorithm will drop significantly. The convergence rate of TSCP in the simulation of reference [22] is only 50%, and Example 1 in Section 6 also shows that the convergence rate is only 22%. Therefore, it is necessary to introduce a better method to deal with complex obstacles.

A flow chart of the proposed HNSCP algorithm is shown in Figure 2.

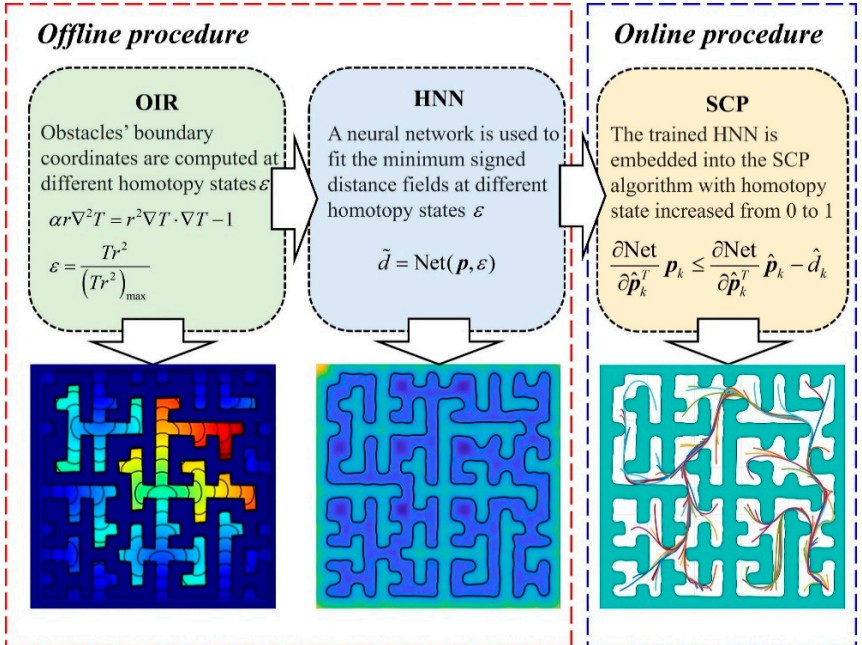

**Figure 2.** Schematic diagram of proposed HNSCP algorithm.

For the convenience of explanation, Figure 2 takes the "maze" in Section 6 as an example. First, mature modeling software (SolidWorks, Dassault Systemes, the United States; AutoCAD, Autodesk, the United States) was used to build the maze and obtain two boundaries, $\partial\widetilde{\Omega}$ and $\partial\Omega$.

Given $\partial\widetilde{\Omega}$ and $\partial\Omega$, the OIR approach proposed in Section 5 calculates the position coordinates of the obstacle boundary in different "growth" states from $\partial\widetilde{\Omega}$ to $\partial\Omega$ with a constant growth speed. We denoted the "growth" state of the obstacle with $\varepsilon \in [0,1]$, also known as the homotopy parameter. The obstacle boundary, cavity, and obstacle at homotopy parameter $\varepsilon$ are denoted as $\partial\Omega^\varepsilon$, $\Omega_v^\varepsilon$, and $\Omega_o^\varepsilon$, respectively. The OIR will output $\partial\Omega^\varepsilon$, $\Omega_v^\varepsilon$, and $\Omega_o^\varepsilon$ under a series of $\varepsilon$ values. The schematic diagram is shown in the lower-

left corner of Figure 2. The black lines are the contours of $\varepsilon$, representing the $\partial\Omega^\varepsilon$ when the homotopy parameter is $\varepsilon$. Areas with a reddish color suggest that $\varepsilon$ is close to 1. Areas with a bluish color indicate that $\varepsilon$ is close to 0.

Without a loss of generality, cavity $\Omega_v^\varepsilon$ is a nonconvex area. The MSD was defined as $d^\varepsilon$.

$$
\begin{aligned}
d^\varepsilon &= d^{\varepsilon+} - d^{\varepsilon-} \quad k = 1, 2, \ldots, N \\
d^{\varepsilon+} &= \inf\{\|\boldsymbol{d}\| \,|\, \boldsymbol{p} + \boldsymbol{d} \in \Omega_v^\varepsilon\} \\
d^{\varepsilon-} &= \inf\{\|\boldsymbol{d}\| \,|\, \boldsymbol{p} + \boldsymbol{d} \notin \Omega_v^\varepsilon\}
\end{aligned}
\tag{6}
$$

The absolute value of $d^\varepsilon$ is the shortest distance from the given position $\boldsymbol{p}$ to the obstacle boundary. The MSD is negative if the coordinates are inside the cavity ($\boldsymbol{p} \in \Omega_v^\varepsilon$), while the MSD is positive if the coordinates are inside the obstacle ($\boldsymbol{p} \in \Omega_o^\varepsilon$).

Subsequently, the HNN was trained using the sample generation and training method proposed in Section 5. The inputs of HNN are the position coordinates and the homotopy parameter $\varepsilon$, and the output is the estimated value of the MSD $\widetilde{d} = \text{Net}(\boldsymbol{p}, \varepsilon)$. The superscript "~" indicates that the value is the prediction. The schematic diagram of the fitting result is shown at the bottom of Figure 2. It is worth noting that Figure 2 only shows the fitting result at $\varepsilon = 1$. The black line is the contour line that satisfies $\text{Net}(\boldsymbol{p}, 1) = 0$, the yellowish color indicates that the predicted $d$ is larger, and the blue color indicates that the predicted $d$ is smaller. Finally, the neural network function $\text{Net}(\boldsymbol{p}, \varepsilon)$, which can predict the MSD with any homotopy parameter, was obtained.

Then, the trained HNN was embedded into SCP, proposed in Section 4, as OACs. A different $\varepsilon$ was selected in the iteration to reduce the difficulty of solving and improve the convergence. In the first iteration, $\varepsilon$ was set to 0. $\varepsilon$ increased linearly with a step size of $\Delta\varepsilon$ as the number of iterations increased, until $\varepsilon$ was equal to 1. The bottom-right corner of Figure 2 shows the trajectory optimization results for different initial conditions.

## 4. Online Procedure

### 4.1. Discretization and Convexification

4.1.1. Discretization

The whole flight process was divided into $N - 1$ segments according to equal time intervals; then, the discrete time step was as follows:

$$
\Delta t = \frac{t_f - t_0}{N - 1}
\tag{7}
$$

Without a loss of generality, take $t_0 = 0$. Given the constant discrete number $N$, the discrete step size $\Delta t$ is only related to the terminal time $t_f$. To address the free final time problem, $\Delta t$ was regarded as a decision variable in this study. The discrete state variables and control variables with serial number $k$ are defined as:

$$
\begin{aligned}
\boldsymbol{p}_k &= \boldsymbol{p}(k\Delta t - \Delta t) \\
v_k &= v(k\Delta t - \Delta t) \\
\boldsymbol{u}_k &= \boldsymbol{u}(k\Delta t - \Delta t) \\
k &= 1, 2, \ldots, N
\end{aligned}
\tag{8}
$$

To avoid the "artificial infeasibility" in the first few iterations, the virtual control variables $\boldsymbol{a}_{vk} \in R^3$ were augmented to control the variables [9]. Using the trapezoidal discrete method, Equation (1) can be discretized and written as follows:

$$
\begin{aligned}
\boldsymbol{p}_{k+1} - \boldsymbol{p}_k - \tfrac{1}{2}\Delta t\, v_k - \tfrac{1}{2}\Delta t\, v_{k+1} &= \mathbf{0} \\
v_{k+1} - v_k - \tfrac{1}{2}\Delta t\, \boldsymbol{u}_k - \tfrac{1}{2}\Delta t\, \boldsymbol{u}_{k+1} - \tfrac{1}{2}\Delta t\, \boldsymbol{a}_{vk} - \tfrac{1}{2}\Delta t\, \boldsymbol{a}_{v(k+1)} - \Delta t\, \boldsymbol{g} &= 0 \\
k &= 1, 2, \ldots, N-1
\end{aligned}
\tag{9}
$$

### 4.1.2. Convexification of Dynamical Equations

Equation (1) was converted into $6\,(N-1)$ algebraic equations. The linearized equations can be obtained by taking the first-order approximation of the Taylor expansion. The coefficient matrices can be simplified as follows:

$$A_k \begin{bmatrix} p_k \\ p_{k+1} \end{bmatrix} + B_k \begin{bmatrix} v_k \\ v_{k+1} \end{bmatrix} + C_k \begin{bmatrix} u_k \\ u_{k+1} \end{bmatrix} + D_k \begin{bmatrix} a_{vk} \\ a_{v(k+1)} \end{bmatrix} + E_k \Delta t = F_k \quad k = 1, 2, \dots, N-1$$

$$A_k = \begin{bmatrix} -I_{3\times3} & I_{3\times3} \\ 0_{3\times3} & 0_{3\times3} \end{bmatrix} B_k = \begin{bmatrix} -\frac{1}{2}\Delta\hat{t}\,I_{3\times3} & -\frac{1}{2}\Delta\hat{t}\,I_{3\times3} \\ -I_{3\times3} & I_{3\times3} \end{bmatrix}$$

$$C_k = \begin{bmatrix} 0_{3\times3} & 0_{3\times3} \\ -\frac{1}{2}\Delta\hat{t}\,I_{3\times3} & -\frac{1}{2}\Delta\hat{t}\,I_{3\times3} \end{bmatrix} D_k = \begin{bmatrix} 0_{3\times3} & 0_{3\times3} \\ -\frac{1}{2}\Delta\hat{t}\,I_{3\times3} & -\frac{1}{2}\Delta\hat{t}\,I_{3\times3} \end{bmatrix} \tag{10}$$

$$E_k = \begin{bmatrix} -\frac{1}{2}(\hat{v}_k + \hat{v}_{k+1}) \\ -\frac{1}{2}\left(\hat{u}_k + \hat{u}_{k+1} + \hat{a}_{vk} + \hat{a}_{v(k+1)} + 2g\right) \end{bmatrix} F_k = \begin{bmatrix} -\frac{1}{2}(\hat{v}_k + \hat{v}_{k+1})\Delta\hat{t} \\ -\frac{1}{2}\left(\hat{u}_k + \hat{u}_{k+1} + \hat{a}_{vk} + \hat{a}_{v(k+1)}\right)\Delta\hat{t} \end{bmatrix}$$

where the variables with the superscript "^" represent reference variables and are regarded as known variables. All coefficient matrices in (10) are constant, and all constraints are converted into affine constraints.

The transformation from (9) to (10) introduces linearized approximation. Trust region constraints were added to ensure that the linearized constraints were still valid [9]. The new variables $\sigma_{uk}$ and $\sigma_{\Delta t}$ were introduced to represent the trust regions of the control variables and the discrete time step, respectively:

$$\|u_k - \hat{u}_k\| \le \sigma_{uk} \quad k = 1, 2, \dots, N \tag{11}$$

$$\|\Delta t - \Delta\hat{t}\| \le \sigma_{\Delta t} \tag{12}$$

The above trust regions were augmented into the cost function in the form of penalty terms. In addition, it is crucial to introduce the new variables $\sigma_{ak}$ and augment them to the cost function in the form of penalty terms to constrain the magnitude of virtual control:

$$\|a_{vk}\| \le \sigma_{ak} \quad k = 1, 2, \dots, N \tag{13}$$

### 4.1.3. Convexification of Obstacle Avoidance Constraints

According to (6), the OACs are then denoted by nonlinear inequations:

$$d_k^{\varepsilon} \le 0 \quad k = 1, 2, \dots, N \tag{14}$$

where $d_k^{\varepsilon}$ is only related to the position coordinate $p_k$ when cavity $\Omega_v^{\varepsilon}$ is determined. Replacing $d_k^{\varepsilon}$ with the HNN leads to the following inequality:

$$\text{Net}(p_k, \varepsilon) \le 0 \quad k = 1, 2, \dots, N \tag{15}$$

Since the HNN is continuously differentiable, Equation (15) can be simplified by using the Taylor expansion, and only the first-order linear terms are retained:

$$\left[\frac{\partial}{\partial \hat{p}_k^T}\text{Net}(\hat{p}_k, \varepsilon)\right] p_k \le \left[\frac{\partial}{\partial \hat{p}_k^T}\text{Net}(\hat{p}_k, \varepsilon)\right] \hat{p}_k - \text{Net}(\hat{p}_k, \varepsilon) - d_{\text{mar}} \quad k = 1, 2, \dots, N \tag{16}$$

One can obtain $\hat{p}_k$ from the result of the previous iteration. The collision safety margin $d_{\text{mar}} \ge 0$ was embedded, which avoids the increased risk of collision due to fitting errors of the HNN. The numerical finite difference method was adopted to approximate the differential operation. It is worth noting that the method proposed in this study cannot guarantee obstacle avoidance between discrete time steps. However, the numerical simulation shows that in most cases, there were only finite isolated points on the trajectory making the constraints (16) tight, and there is the possibility of violating the constraint

near these isolated points. We addressed this problem by choosing the appropriate discrete number $N$ and safety margin $d_{\mathrm{mar}}$. For Example 1, the simulation shows that the OACs are satisfied between discrete time steps when $N \geq 30$ and $d_{\mathrm{mar}} \geq 3$ m. For Example 2, it is suitable to choose $N \geq 100$ and $d_{\mathrm{mar}} \geq 5$ m.

### 4.1.4. Convexification of Control Magnitude Constraints

When the lower bound of the thrust acceleration $u_{\min}$ is non-zero, the feasible set of the thrust acceleration is non-convex. New slack variables were introduced to transform the thrust magnitude constraints (2) in the primal problem into convex ones:

$$\|\boldsymbol{u}_k\| \leq \Gamma_k \tag{17}$$

$$u_{\min} \leq \Gamma_k \leq u_{\max} \tag{18}$$

where $\Gamma_k$ represents the slack variable. Reference [36] and a large number of numerical simulations show that although the feasible region of the problem has changed after relaxation, the optimal solutions are equivalent.

### 4.1.5. Convexification of Cost Function

After augmenting the penalty terms, the modified linear cost function is listed as follows:

$$J = \sum_{k=1}^{N} \left( \hat{\Gamma}_k \Delta t + \Delta \hat{t}\, \Gamma_k + \omega_a \sigma_{ak} + \omega_u \sigma_{uk} \right) + \omega_{\Delta t} \sigma_{\Delta t} \tag{19}$$

where $\omega_a$, $\omega_u$, and $\omega_{\Delta t}$ are fixed penalty coefficients that can be set manually by users. The selection of these parameters follows two principles. First, $\omega_a$ needs to be a large positive number to ensure that the magnitude of the virtual control variable converges to zero. Second, $\omega_u$ and $\omega_{\Delta t}$ need to be selected as small positive numbers to ensure the convergent solution is near the reference one.

The convexified problem is summarized in Problem 2. Problem 2 is an SOCP and can be solved efficiently using the prime-dual interior point method.

Problem 2:

Cost function: $\underset{\Delta t, \boldsymbol{p}_k, v_k, \boldsymbol{u}_k, \boldsymbol{a}_{vk}, \sigma_{uk}, \sigma_{ak}, \sigma_{\Delta t}, \Gamma_k}{\text{minimize}}$ (19)

Subject to: (4), (10–13), (16–18) with given $\varepsilon$

Table 1 lists the number of variables, affine equations, affine inequalities, and second-order cones.

**Table 1.** Variables and constraints in Problem 2.

| Category | Item | Number | Dimension of the Cone | Total Number |
|---|---|---|---|---|
| Variables to be optimized | Time step $\Delta t$ | 1 | - | $15\,N + 2$ |
| | Position $\boldsymbol{p}_k$ | $3\,N$ | | |
| | Velocity $v_k$ | $3\,N$ | | |
| | Control $\boldsymbol{u}_k$ | $3\,N$ | | |
| | Virtual control $\boldsymbol{a}_{vk}$ | $3\,N$ | | |
| | Control trust regions $\sigma_{uk}$ | $N$ | | |
| | Virtual control magnitude relaxation variables $\sigma_{ak}$ | $N$ | | |
| | Time trust region $\sigma_{\Delta t}$ | 1 | | |
| | Control magnitude relaxation variables $\Gamma_k$ | $N$ | | |
| Affine equality constraints | Boundary value conditions, Equation (4) | 12 | - | $6\,N + 6$ |
| | Dynamic constraints, Equation (10) | $6\,(N-1)$ | | |
| Affine inequality constraints | OACs, Equation (16) | $N$ | - | $3\,N$ |
| | Control magnitude constraints, Equation (18) | $2\,N$ | | |
| Second order cone constraints | Control trust region constraints, Equation (11) | $N$ | 4 | $3\,N + 1$ |
| | Time trust region constraint, Equation (12) | 1 | 2 | |
| | Virtual control constraints, Equation (13) | $N$ | 4 | |
| | Control magnitude constraints, Equation (17) | $N$ | 4 | |

Assume $N = 100$. According to Table 1, there are 1502 variables, 606 affine equations, 300 affine inequations, and 301 s-order cones. Therefore, Problem 2 is a medium-scale SOCP with thousands of variables.

### 4.2. Successive Iterative Algorithm

The successive iterative algorithm solving the primal problem is shown below.

Algorithm 1 required an initial reference trajectory. Unfortunately, it is difficult to generate high-quality initial reference trajectories in real time due to the complexity and variability of the problem [1]. In this work, Algorithm 2 was adopted to generate rough initial reference trajectories efficiently.

---

**Algorithm 1**: Online Procedure of Homotopy Network Sequential Convex Programming (HNSCP)

---

**Step 1:** Algorithm initialization. Set the maximum number of iterations $i_{\max}$. Set the penalty coefficient $\omega_a$, $\omega_u$ and $\omega_{\Delta t}$. Set the iteration convergence criterion $\varepsilon_{\max,u}$, $\varepsilon_{\max,t}$, $\varepsilon_{\max,a}$. Set the increment step of the homotopy variable $\Delta \varepsilon$. Set the homotopy variable $\varepsilon = 0$;
**Step 2:** Set initial reference variables, $\Delta \hat{t}^{(0)}$, $\hat{p}_k^{(0)}$, $\hat{v}_k^{(0)}$, $\hat{u}_k^{(0)}$, $\hat{a}_{vk}^{(0)}$, according to Algorithm 2;
For $i = 0$: $i_{\max}$
    If $\varepsilon < 1.0$
        $\varepsilon = \varepsilon + \Delta \varepsilon$
    Else
        $\varepsilon = 1.0$
    End
**Step 3:** Solve Problem 2, and the obtained solution is used to update the reference variables $\Delta \hat{t}^{(i+1)}$, $\hat{p}_k^{(i+1)}$, $\hat{v}_k^{(i+1)}$, $\hat{u}_k^{(i+1)}$, $\hat{a}_{vk}^{(i+1)}$;
    If $\sum_{k=1}^{N} \sigma_{uk}^{(i)} \leq \varepsilon_{\max,u}$, $\sigma_{\Delta t}^{(i)} \leq \varepsilon_{\max,t}$, $\sum_{k=1}^{N} \sigma_{ak}^{(i)} \leq \varepsilon_{\max,a}$ and $\varepsilon = 1.0$
        A convergent solution $\Delta \hat{t}^{(i)}$, $\hat{p}_k^{(i)}$, $\hat{v}_k^{(i)}$, $\hat{u}_k^{(i)}$, $\hat{a}_{vk}^{(i)}$ is obtained;
        Return;
    End
End
**Step 4:** Fail to converge;
Return;

---

---

**Algorithm 2:** Initialization of HNSCP's Reference Variables

---

**Step 1:** Obtain the initial states and the terminal states $p_0$, $v_0$, $p_f$, $v_f$
**Step 2:** Set the reference speed constant $v_{\text{ref}}$;
**Step 3:** Set the linear reference variables;
$\Delta \hat{t}^{(0)} = \|p_f - p_0\|/v_{\text{ref}}$;
For $k = 1$: $N$
    $\hat{p}_k^{(0)} = p_0 + (k-1)(p_f - p_0)/(N-1)$; $\hat{v}_k^{(0)} = (p_f - p_0)/\Delta \hat{t}^{(0)}$; $\hat{u}_k^{(0)} = [0, u_{\max}, 0]^T$;
    $\hat{a}_{vk}^{(0)} = [0, 0, 0]^T$;
End
Return;

---

## 5. Offline Procedure

### 5.1. Obstacle Interface Regression Approach

The eikonal equation, a typical hyperbolic partial differential equation, was first applied to wave propagation. The position of the wavefront at different time steps can be obtained by solving the partial differential equation. Inspired by this, this section extends the method to describe the growth and construction of obstacles and proposes a universal method to address this issue. There are mature methods and commercial software for solving boundary value problems of partial differential equations. Therefore, the proposed method enjoys simple programming and strong universality. Consequently, it is more competitive than other methods.

#### 5.1.1. Basic Principles

The obstacle's "growth" process is a kind of interface regression phenomenon. This phenomenon has been extensively studied in optics [37], combustion [38,39], and image processing [40]. Sethian [41,42] proposed the level-set method and the fast marching method. These two methods solve the initial value problem of the level-set equation and the boundary value problem of the eikonal equation, respectively. The solution describes the interface at the given regression state. Although these two methods have been widely used, there are still many difficulties in solving hyperbolic partial differential equations. Mokrý [37] and Li [39] proposed a method to solve an ellipse-form eikonal equation, which

can be directly embedded in the finite element software. The original eikonal equation is as follows:

$$\nabla T(\boldsymbol{p}) \cdot \nabla T(\boldsymbol{p}) = \frac{1}{r(\boldsymbol{p})^2} \tag{20}$$

where $T$ is a scalar field. Its physical meaning is the moment when the interface propagates to the given position, so $T$'s contour line is the interface. $r$ is the regression speed.

It is hard to obtain a reliable solution for (20) by using existing numerical methods, so (20) needs to be approximated to an ellipse form:

$$\alpha r \nabla^2 T = r^2 \nabla T \cdot \nabla T - 1 \tag{21}$$

where $\alpha$, a small-enough positive number, is the diffusion term coefficient set manually by users. It is worth noting that (21) degenerates into (20) if $\alpha = 0$.

Equation (21) is a nonlinear Poisson equation. Solving (21) requires a series of boundary conditions. The initial interface adopts the Dirichlet condition $T = 0$, and the rest of the boundaries adopt the "zero flux" condition. The nonlinear Poisson equation can then be transformed into sparse quadratic equations using a finite element method. The system of algebraic equations can be efficiently solved using the Newton–Raphson method. Powerful commercial finite element software (COMSOL Multiphysics, Comsol Inc, Zürich, Switzerland; ANSYS, Ansys Inc, Champaign, IL, USA) can be used for mesh generation, equation resolving, and post-processing.

It is worth noting that the proposed method can only solve the approximate solution. The value of $\alpha$ determines the performance of the algorithm. If $\alpha$ is large, the approximate error of (21) is large, which will eventually lead to a large error in the regression result. If $\alpha$ is small, it will be difficult for the Newton–Raphson method to converge. To take into account the convergence and error, with the aid of a large number of numerical simulations, a reasonable range of $\alpha$ was concluded [39]:

$$\alpha \in [0.1h,\ 0.5h] \tag{22}$$

where $h$ is the scale of the local mesh element.

### 5.1.2. Implementation

Firstly, the Dirichlet boundary condition was set as:

$$T(\boldsymbol{p}) = 0, \quad \boldsymbol{p} \in \partial\widetilde{\Omega} \tag{23}$$

The regression speed $r$ was used to describe spatial distribution of the obstacle:

$$r(\boldsymbol{p}) = \begin{cases} \kappa r_0 & \boldsymbol{p} \notin \Omega_{\mathrm{o}} \\ r_0 & \boldsymbol{p} \in \Omega_{\mathrm{o}} \end{cases} \tag{24}$$

where $\kappa$ is a number much less than 1 and $r_0$ is the unit regression speed, which can be taken as 1 m/s. Its practical implication is as follows: the regression speed of the interface in the obstacle $\Omega_{\mathrm{o}}$ is $r_0$, while the regression speed in the cavity $\Omega_{\mathrm{v}}$ is so small that it can be ignored numerically. Since the interface will preferentially pass through the obstacle region $\Omega_{\mathrm{o}}$, the $T$ contours describe the "growth" boundaries of the obstacle from $\partial\widetilde{\Omega}$ to the given $\partial\Omega$, as shown in Figure 3.

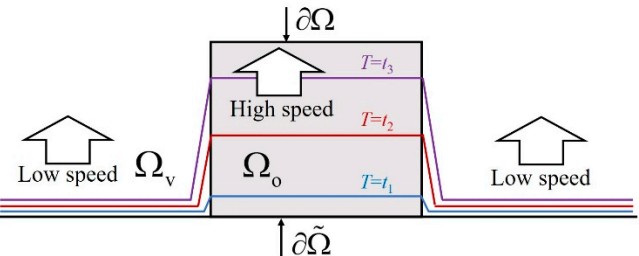

**Figure 3.** Schematic diagram of obstacle interface regression.

The normalized scalar field $\varepsilon$ was defined as:

$$\varepsilon = \frac{Tr^2}{(Tr^2)_{\max}} \tag{25}$$

The subscript "max" means obtaining the maximum value of the whole field, and the $\varepsilon$ field is a scalar field ranging between 0 and 1. The $\varepsilon$ field's contours represent the obstacles' boundaries in different "growth" states. For example, the contour line $\varepsilon = 0$ indicates no obstacles, the contour line $\varepsilon = 0.45$ suggests that the obstacle has grown to 45%, and the contour line $\varepsilon = 1$ indicates that the obstacle has been restored to $\Omega_o$.

Figure 4 shows the flow charts and results for the two examples in Section 6, respectively. First, finite element software or CAD software was used to model the obstacle; the blue area represents the cavity, and the yellow area represents the obstacle. Then, the interface regression speed was set according to (24), and the boundary conditions were set according to (23). After dividing the uniform triangular second-order element or the tetrahedral second-order element, the Newton iteration method was used to solve (21) to obtain the $T$ field. Finally, the $\varepsilon$ field was obtained according to (25). The method can return the "growth" boundary coordinates for a given $\varepsilon$, laying the foundation for the sample generation of the HNN in the next section.

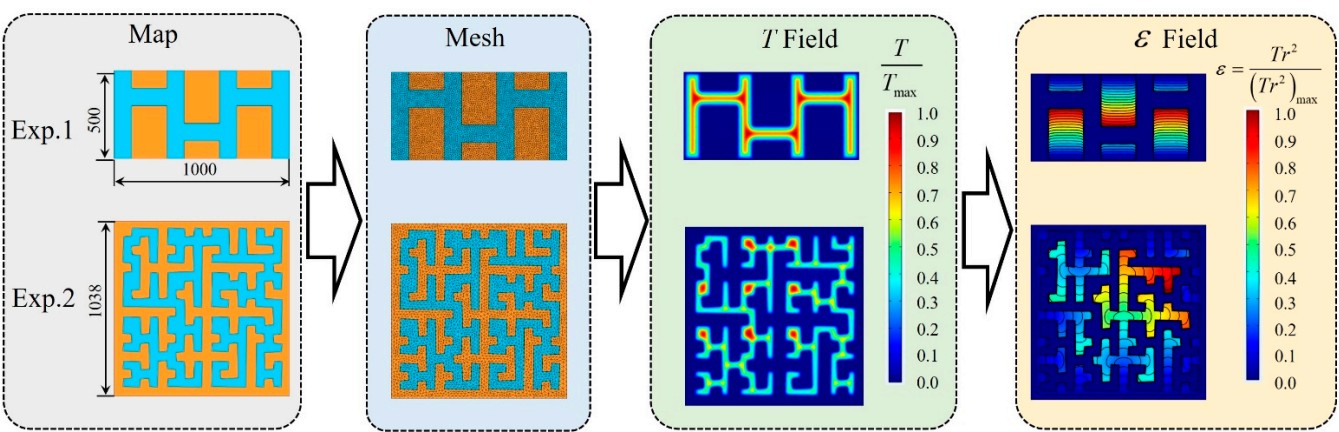

**Figure 4.** OIR flow charts of two examples in Section 6.

*5.2. Homotopy Neural Network*

5.2.1. Structure

The feedforward neural network (FNN) was the first artificial neural network invented. The units in each layer can receive signals from the units in the previous layer and transmit signals to the next layer. According to the universal approximation theorem, for an FNN with a linear output layer and at least one hidden layer using a nonlinear activation function, as long as the number of units in the hidden layer is large enough, the FNN can approximate any real function defined in a bounded closed set [43,44].

Considering the training time cost and fitting accuracy, an FNN composed of three hidden layers was used to predict the MSD field. The input layer was denoted as layer 0,

the hidden layers were denoted as layers 1 to 3, and the output layer was denoted as layer 4. Table 2 gives the relevant notation used in this section.

**Table 2.** The meaning of the associated notation.

| Notation | Explanation |
|---|---|
| $M_l$ | Number of units in layer $l$ |
| $f_l(\cdot)$ | Activation function of the units in layer $l$ |
| $\boldsymbol{W}^{(l)} \in R^{M_l \times M_{l-1}}$ | Weight matrix from layer $l - 1$ to layer $l$ |
| $\boldsymbol{b}^{(l)} \in R^{M_l}$ | Bias vector from layer $l - 1$ to layer $l$ |
| $\boldsymbol{z}^{(l)} \in R^{M_l}$ | Net input (net activation) of units in layer $l$ |
| $\boldsymbol{a}^{(l)} \in R^{M_l}$ | Output (activation) of units in layer $l$ |

Setting the input of the neural network as the position coordinates and homotopy parameters, the following equation exists:

$$\boldsymbol{a}^{(0)} = \left(\boldsymbol{p}^T, \varepsilon\right)^T = \left(p_x, p_y, p_z, \varepsilon\right)^T \tag{26}$$

Then, the output of the other layers can be calculated using the following propagation formula:

$$\begin{aligned} \boldsymbol{z}^{(l)} &= \boldsymbol{W}^{(l)}\boldsymbol{a}^{(l-1)} + \boldsymbol{b}^{(l)} \\ \boldsymbol{a}^{(l)} &= f_l(\boldsymbol{z}^{(l)}) \end{aligned} \quad l = 1, 2, 3, 4 \tag{27}$$

The neural network will recursively obtain the output according to the following process. First, the net activation of layer $l - 1$ is calculated from the activation of layer $l - 1$ and the weight matrix and bias vector of layer $l$. The nonlinear activation function is used to calculate the activation of the $l$ layer. This process is repeated until reaching the output layer. The activation of the output layer satisfies the following equation:

$$\boldsymbol{a}^{(4)} = \widetilde{d} \tag{28}$$

where $d$ represents the MSD and its superscript "~" indicates that the value is the prediction. The schematic diagram of the HNN's structure is shown in Figure 5.

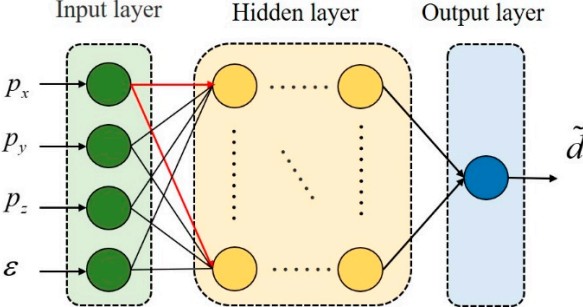

**Figure 5.** Structure diagram of the HNN.

Generally speaking, to reduce the fitting error, the number of units in each layer should be appropriately increased if the obstacles are more complex. The two examples in Section 6 used different unit scales to obtain better performance, and the specific parameters are shown in Table 3. It is worth noting that the activation function was selected similarly. The hidden layers adopted the sigmoid activation function, and the output layer adopted the linear activation function.

**Table 3.** Selection of the HNN structure in different examples.

| Parameter | Example 1 | Example 2 |
|-----------|-----------|-----------|
| $M_0$ | 4 | 4 |
| $M_1$ | 10 | 40 |
| $M_2$ | 20 | 40 |
| $M_3$ | 10 | 40 |
| $M_4$ | 1 | 1 |

5.2.2. Sample Generation

The traditional method of fitting control variables and trajectories using supervised learning requires many samples [28–33], and the only way to obtain samples is to solve complex optimal control problems. Therefore, acquiring a large number of samples has become a challenge in neural network training. Training the HNN proposed in this study only required calculating the MSD. Therefore, a large number of samples can be generated efficiently. The specific algorithm is given below.

5.2.3. Training

First, a loss function called the mean square error (MSE) was defined to evaluate the fitting accuracy of the HNN:

$$\sigma_{\mathrm{MSE}} = \frac{\sum\limits_{i=1}^{n}\left(d_i - \tilde{d}_i\right)^2}{n\max\left(d_i^2\right)} = \frac{\sum\limits_{i=1}^{n}\left[d_i - \mathrm{Net}\left(\boldsymbol{p}_i^T, \varepsilon_i\right)\right]^2}{n\max\left(d_i^2\right)} \tag{29}$$

where $n$ is the total number of samples, $d_i$ is the target value generated by Algorithm 3, and $\tilde{d}_i$ is the estimated value of the HNN. A total of 85% of the data set was used for training and 15% was used for test. The Bayesian regularization training method [45] was used to train the HNN. This method can effectively avoid overfitting and has a better fitting performance for complex problems. The training results of the two examples in Section 6 are shown in Table 4.

---

**Algorithm 3:** Sample Generation

---

Step 1: Obtain the output of the OIR algorithm: $\partial\Omega^\varepsilon$, $\Omega_\mathrm{v}^\varepsilon$ and $\Omega_\mathrm{o}^\varepsilon$.

Step 2: Generate $n$ random samples $\left(\boldsymbol{p}_i^T, \varepsilon_i\right)^T$, where $\boldsymbol{p}_i \in \partial\Omega^\varepsilon \cup \Omega_\mathrm{v}^\varepsilon \cup \Omega_\mathrm{o}^\varepsilon$ and $\varepsilon_i \in [0, 1]$;

For $i = 1$: $n$

    Step 3: Compute the MSD according to (6);

    Step 4: Store the data in the sample set;

End

Return;

---

**Table 4.** Statistics of training results.

| Parameter | Example 1 | Example 2 |
|-----------|-----------|-----------|
| Number of samples $n$ | 54,270 | 77,920 |
| Mean square error (MSE) | $9.0688 \times 10^{-6}$ | $3.0411 \times 10^{-5}$ |
| MSE of the training set | $8.9248 \times 10^{-6}$ | $2.9356 \times 10^{-5}$ |
| MSE of the test set | $9.8816 \times 10^{-6}$ | $3.6390 \times 10^{-5}$ |
| Training time | 30 min 14 s | 3 h 21 min 10 s |

The MSE of the training set and the sample set in the two examples were close, which indicates that the training did not overfit. In addition, all test set errors were at a small level, showing that the HNN had an excellent fitting performance.

## 6. Numerical Examples

This section presents the application of the algorithm in two scenarios with complex obstacles. It is worth noting that although a 3-DOF dynamic model was established, only the trajectory in the $x - y$ plane was drawn in the results for the convenience of display. All numerical simulations were performed on a PC with an Intel Core i7-11700k at 3.5 GHz and 16 GB of RAM. We adopted ECOS [46] as the interior point solver, and the programming language was C.

### 6.1. Example 1: Comparison with State-of-Art Algorithms

This section compares the results between HNSCP and two other advanced SCP-based algorithms (continuous state-triggered constraint (CSTC) method [21] and homotopy (HOMO) method [4]). The comparison items included real-time, convergence, and optimality performance.

First, the obstacle and cavity were manually set. Then, the OIR method was used to compute the "growth" of the obstacle boundary, and Algorithm 3 was used to generate a large number of samples for HNN training. The fitting performance of the trained HNN is shown in Figure 6. Ultimately, Algorithm 1 and Algorithm 2 can be used to generate the optimal trajectory in real-time.

Figure 6 shows the prediction results of the HNN when the homotopy parameter $\varepsilon$ increased from small to large. It can be seen that with an increase in $\varepsilon$, the MSD field predicted by the HNN changed from simple to complex and gradually approached the MSD field of the primal problem. When $\varepsilon = 1$ (see Figure 6d), the recovery of the MSD field was complete. The vehicle's initial and terminal states were set, and then the HNSCP, HOMO, CSTC, and TSCP algorithms were adopted to solve the optimal trajectory. The specific parameter settings of the algorithm are shown in Table 5, and the results are shown in Table 6 and Figure 7.

It can be seen from Table 6 that under the same parameter settings, HNSCP, HOMO, and CSTC could all obtain the local optimal solution. Among them, the performance index obtained by HNSCP was the smallest, the number of iterations was the smallest, and the elapsed time was also the shortest. Therefore, under these boundary conditions, the HNSCP algorithm was better. It is worth noting that the subproblem was infeasible after six iterations using the TSCP algorithm.

Figure 7a shows the iterative details of the HNSCP algorithm. It was found that in the first few iterations, the trajectory traversed the obstacle. As the number of iterations increased, the trajectory gradually avoided obstacles and converged to a feasible solution. Figure 7b compares the trajectories solved by different algorithms. The CSTC results were quite different from those of the other two algorithms, and the trajectory was more curved, which significantly increased the consumption of fuel. Figure 7c shows the thrust acceleration profile in the "bang-bang" form, which is also the basic feature of the fuel-optimal trajectory. In order to compare the performance of different algorithms more comprehensively, a Monte Carlo simulation was conducted with 100 samples under different discrete number conditions. Given the discrete number, the initial positions of the vehicle were randomly selected to solve the optimal trajectories. The statistical results are shown in Table 7. The optimality performance was evaluated with $I_{\mathrm{opt}}$, and its expression is as follows:

$$I_{\mathrm{opt}} = \sum_{i=1}^{m} \frac{J^*}{m J^*_{\mathrm{HNSCP}}} \tag{30}$$

where $m$ is the number of Monte Carlo test samples. $J^*$ is the velocity increment solved by the algorithm to be evaluated, and $J^*_{\mathrm{HNSCP}}$ is the velocity increment solved by HNSCP.

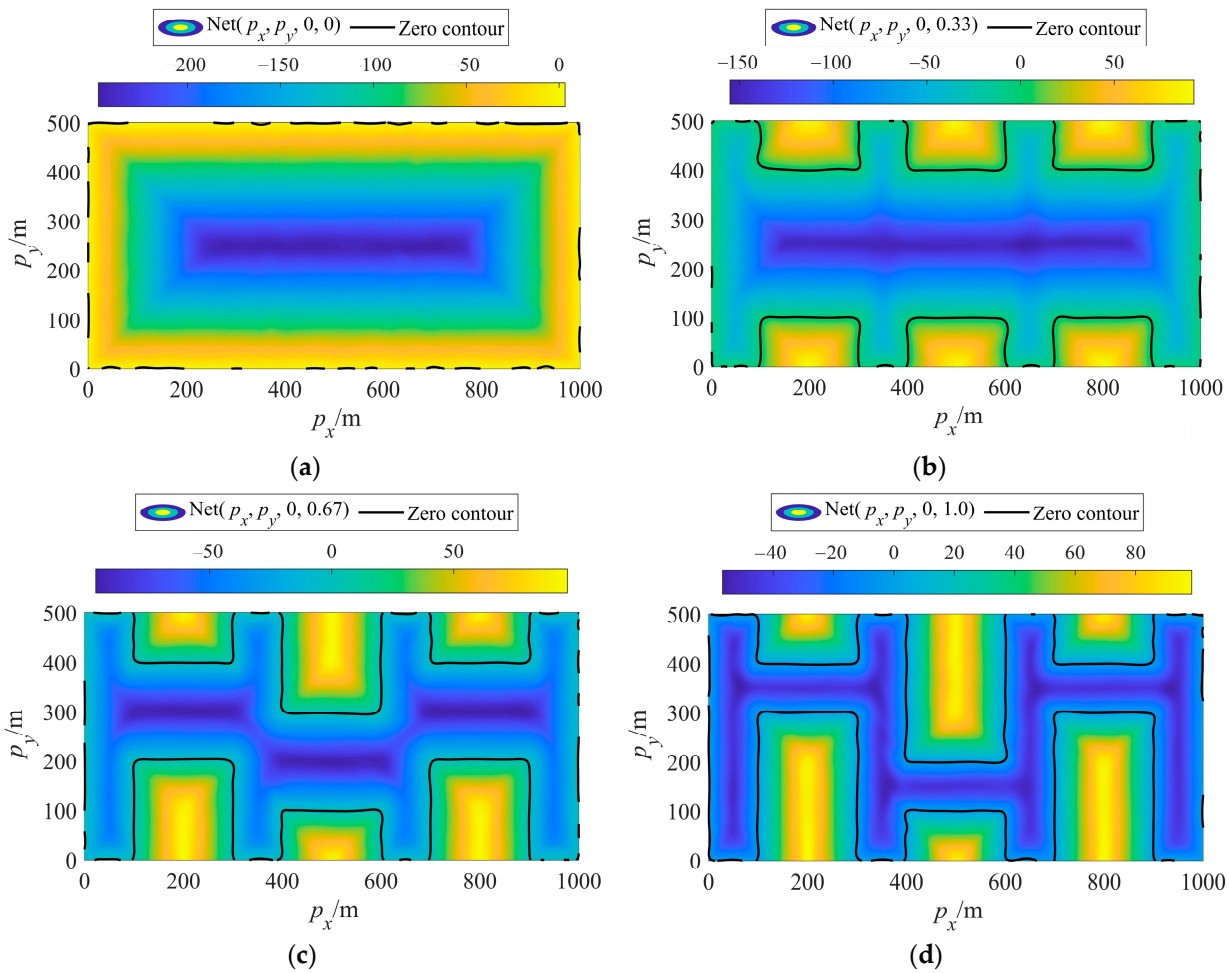

**Figure 6.** Example 1's MSD fields with different homotopy parameters. (**a**) $\varepsilon = 0.0$, (**b**) $\varepsilon = 0.33$, (**c**) $\varepsilon = 0.67$, and (**d**) $\varepsilon = 1.0$.

**Table 5.** Parameter setting of Example 1.

| Parameter | Value | Parameter | Value |
|---|---|---|---|
| $p_0$ | $[50, 10, 0]^T$ m | $\omega_a$ | 100,000.0 |
| $v_0$ | $[0, 0, 0]^T$ m/s | $\omega_u$ | 0.1 |
| $p_f$ | $[950, 10, 0]^T$ m | $\omega_{\Delta t}$ | 0.1 |
| $v_f$ | $[0, 0, 0]^T$ m/s | $\Delta\varepsilon$ | 0.2 |
| $u_{max}$ | 20 m/s$^2$ | $v_{ref}$ | 30 m/s |
| $u_{min}$ | 5 m/s$^2$ | $d_{mar}$ | 5 m |

**Table 6.** Comparison between different SCP-based algorithms ($N = 100$).

| Algorithm | Velocity Increment (m/s) | Number of SCP Iterations | Elapsed Time (s) | Terminal Time (s) |
|---|---|---|---|---|
| HNSCP | 490.80 (Best) | 19 (Best) | 0.487 (Best) | 44.93 |
| HOMO | 509.07 | 19 (Best) | 1.710 | 47.80 |
| CSTC | 609.48 | 29 | 0.920 | 56.35 |
| TSCP | - | Infeasible after 6 | - | - |

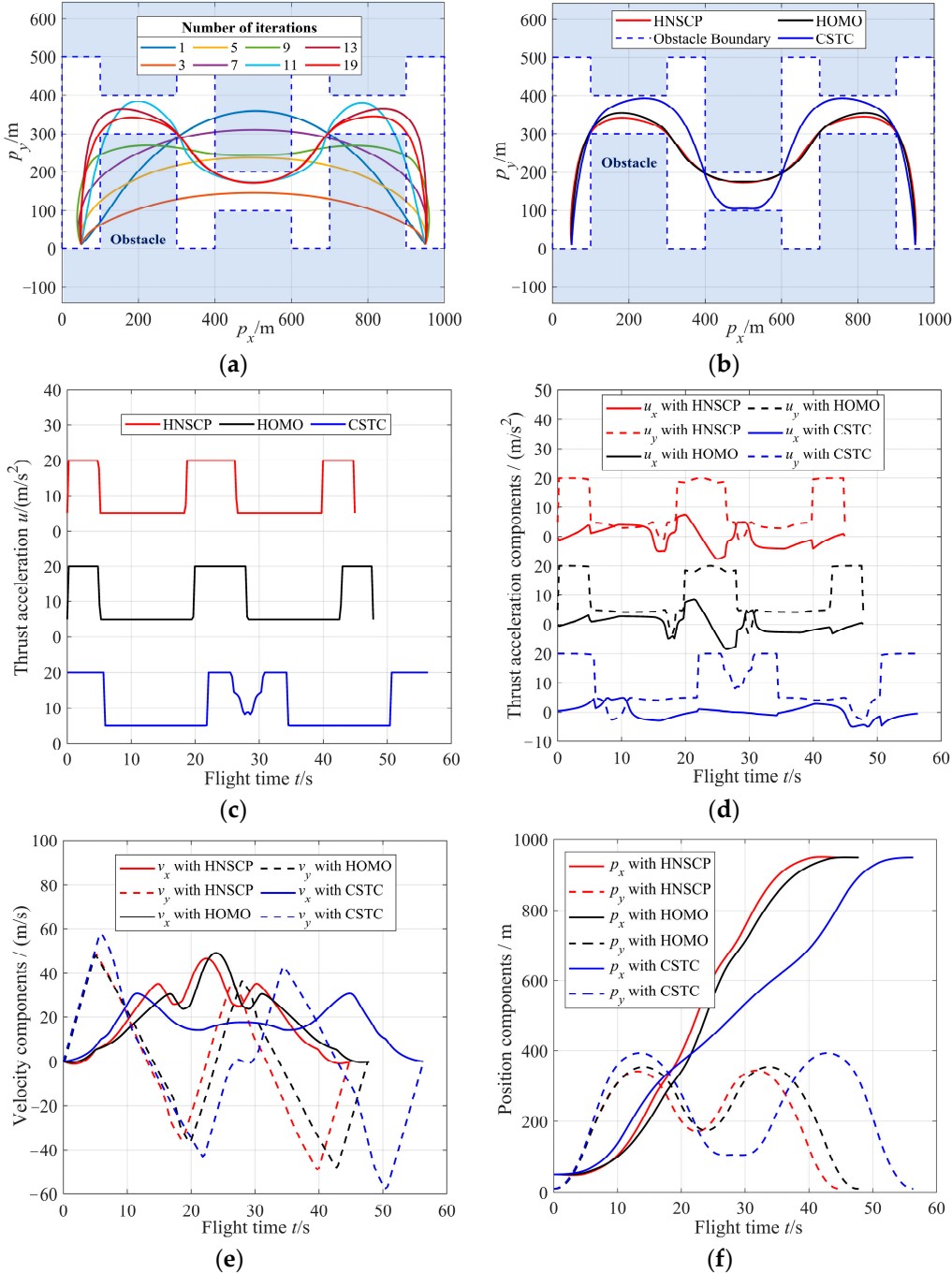

**Figure 7.** The results of Example 1. (**a**) Trajectories obtained by HNSCP algorithm with different iteration numbers. (**b**) Optimal trajectories solved by different algorithms, namely HNSCP, HOMO, and CSTC. (**c**) Profiles of thrust acceleration magnitude. (**d**) Profiles of thrust acceleration components. (**e**) Profiles of velocity components. (**f**) Profiles of position components.

For the convergence rate, the HNSCP algorithm could achieve 100% convergence, which was slightly higher than that of the HOMO and CSTC algorithms and much higher than that of the TSCP algorithm. Therefore, HNSCP has certain advantages in ensuring the safe and reliable flight of the vehicle. For real-time performance, HNSCP's and TSCP's elapsed times were relatively short. The CSTC algorithm needed more iterations. Therefore, the real-time performance was slightly inferior. The HOMO algorithm introduced more auxiliary variables and auxiliary cone constraints, making each iteration longer. Therefore, the real-time performance was the worst. In fact, assuming that the cavity is composed of a

union of $n_{ss}$ cuboid spaces, the HOMO algorithm needs $n_{ss}N$ extra auxiliary variables. As the number of obstacles increases, the scale of the problem will expand dramatically. The relationship between the elapsed time and the discrete number is shown in Figure 8. For optimality, HNSCP, HOMO, and TSCP were similar, while the trajectory planned by the CSTC algorithm was more curved, and the performance index was usually 1.1–1.25 times larger than that of the other algorithms.

**Table 7.** Monte Carlo simulation of Example 1.

| Algorithm | N | Convergence Rate | Average Elapsed Time (s) | Solver's Average Consumed Time (s) | Optimality Performance | Can it be Extended to Exp. 2 |
|---|---|---|---|---|---|---|
| HNSCP | 30 | 100% | 0.1087 | 0.0950 | 1.0 | |
| | 60 | 100% | 0.2546 | 0.2129 | 1.0 | |
| | 100 | 100% | 0.5249 | 0.4124 | 1.0 | Easy |
| | 150 | 100% | 1.0450 | 0.7658 | 1.0 | |
| HOMO (Poor real-time performance) | 30 | 96% | 0.3218 | 0.2782 | 0.9912 | |
| | 60 | 98% | 0.7077 | 0.5451 | 1.0001 | |
| | 100 | 100% | 1.6043 | 1.1096 | 1.0058 | Hard |
| | 150 | 100% | 3.6225 | 2.0957 | 1.0153 | |
| CSTC (Poor optimality performance) | 30 | 99% | 0.1175 | 0.1050 | 1.1052 | |
| | 60 | 99% | 0.3441 | 0.2980 | 1.1559 | |
| | 100 | 100% | 0.8270 | 0.6561 | 1.2239 | Hard |
| | 150 | 98% | 1.6957 | 1.2685 | 1.2217 | |
| TSCP (Poor convergence performance) | 30 | 22% | 0.1093 | 0.0968 | 1.0234 | |
| | 60 | 18% | 0.2488 | 0.2117 | 1.0001 | |
| | 100 | 18% | 0.4781 | 0.3856 | 1.0001 | Easy |
| | 150 | 18% | 0.8242 | 0.6172 | 1.0000 | |

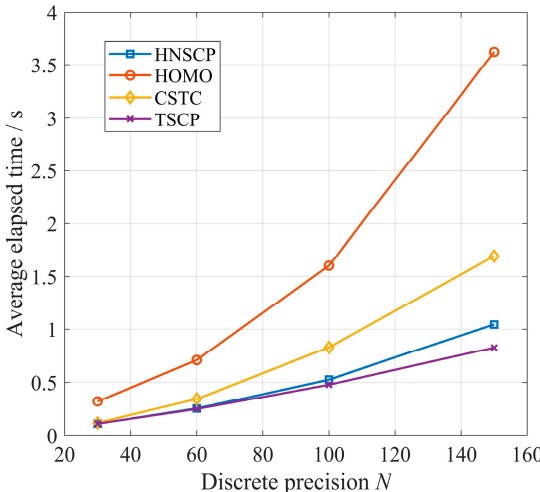

**Figure 8.** The relationship between elapsed time and discrete number.

In summary, each algorithm had some shortcomings. HOMO had poor real-time performance, the CSTC algorithm had a poor performance index, and the TSCP algorithm had poor convergence. However, the HNSCP algorithm could balance various evaluation indicators, and its performance was relatively good. In addition, CSTC and HOMO were also limited because the cavity must be abstracted as a union of multiple cuboids, which cannot well-simulate real and complex environments and has poor extendibility. Conversely, HNSCP can be applied to a wide range of scenarios, which will be shown in Example 2.

### 6.2. Example 2: Application in a Complex Maze

This section increases the obstacles' complexity and verifies the HNSCP algorithm's adaptability. First, a flat "maze" was constructed using modeling software. Then, the steps in Figure 2 were followed to solve the optimal trajectory. The fitting performance of the trained HNN is shown in Figure 9. As the homotopy parameter increased, the signed distance field predicted by the HNN gradually recovered the primal one.

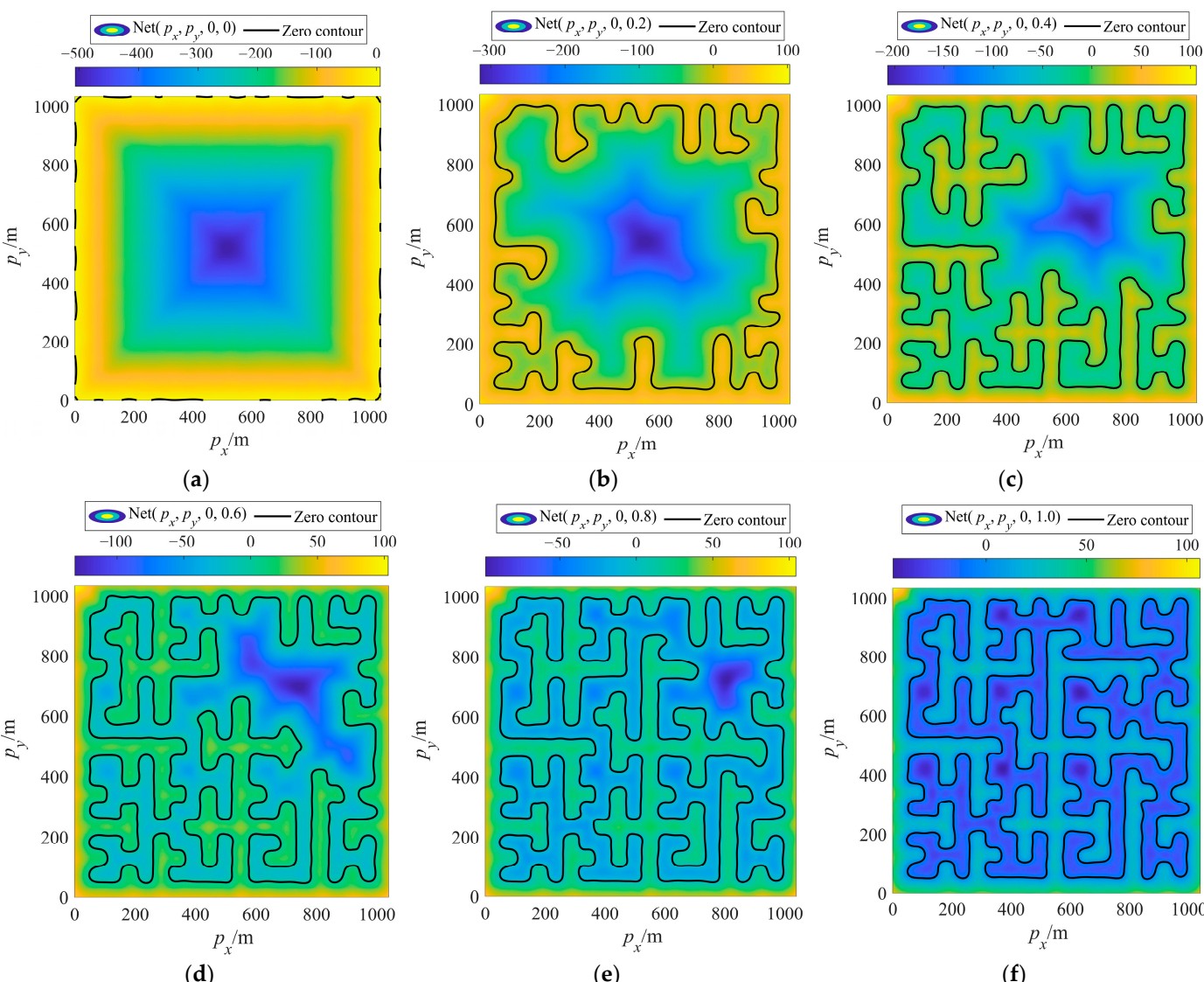

**Figure 9.** Example 2's MSD field with different homotopy parameters. (**a**) $\varepsilon = 0.0$, (**b**) $\varepsilon = 0.2$, (**c**) $\varepsilon = 0.4$, (**d**) $\varepsilon = 0.6$, (**e**) $\varepsilon = 0.8$, and (**f**) $\varepsilon = 1.0$.

The test was divided into a single test and a Monte Carlo test. The settings of the algorithm parameters in a single test were slightly different from those in Table 5. The initial position coordinate was set to $[90, 80, 0]^T$, the terminal coordinate was $[970, 80, 0]^T$, and $\Delta\varepsilon = 0.02$. The other parameters were the same as those in Table 5. The Monte Carlo test was still randomly selecting the initial position coordinates to solve the trajectories. The results obtained for both tests are shown in Table 8 and Figure 10.

**Table 8.** The result of Example 2.

| Single Test | Velocity increment (m/s) | Number of Iterations | Elapsed Time (s) | Terminal Time (s) |
|---|---|---|---|---|
| Results | 751.21 | 59 | 1.728 | 60.179 |

| Monte Carlo Test | Sample Quantity | Convergence Rate | Average Elapsed Time (s) | Solver's Average Consumed Time (s) |
|---|---|---|---|---|
| Results | 100 | 100% | 1.945 | 1.487 |

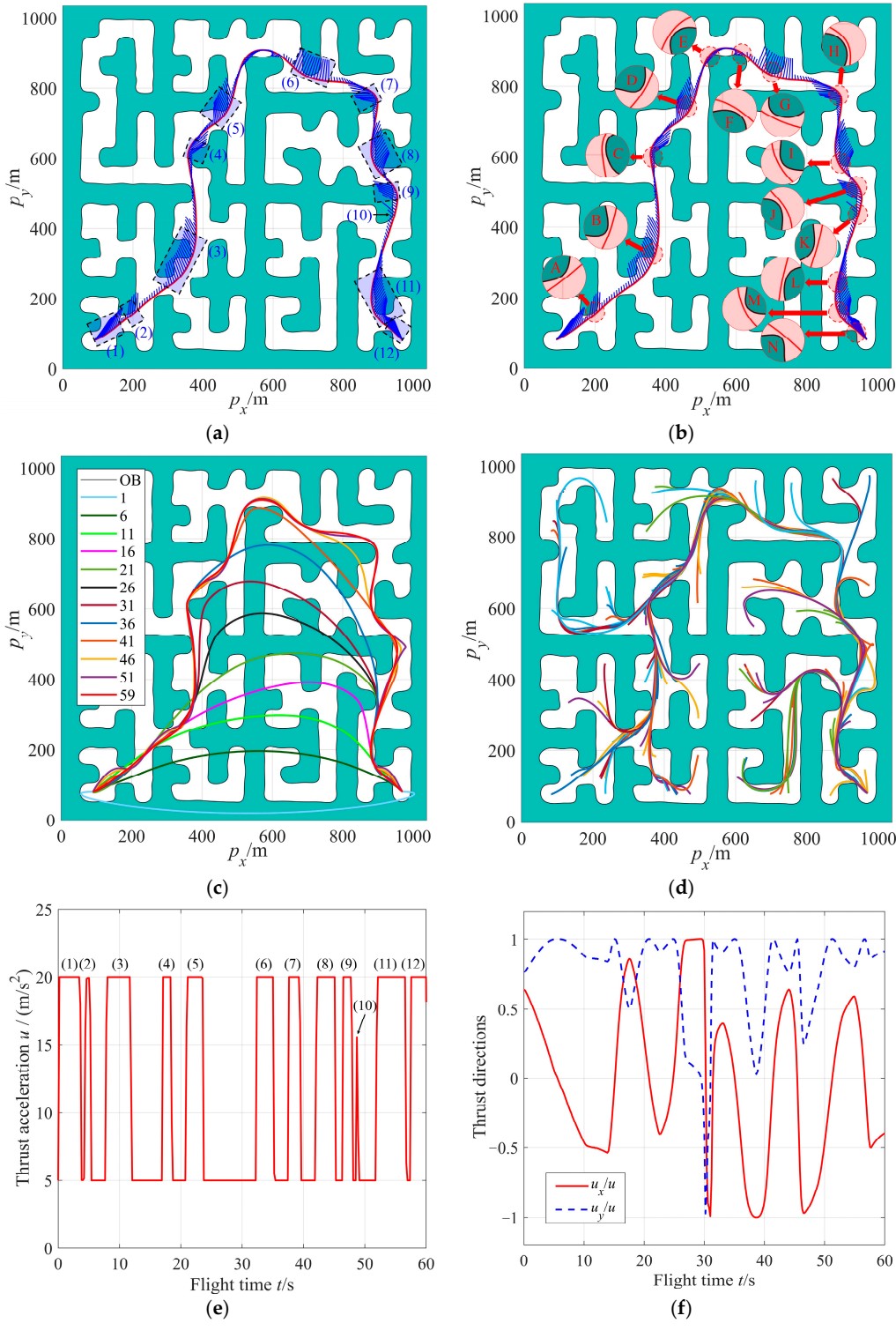

**Figure 10.** *Cont.*

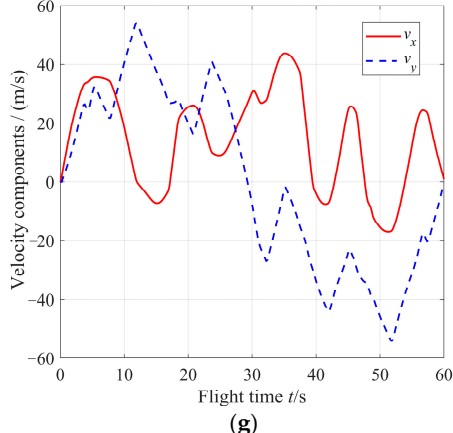

(**g**)

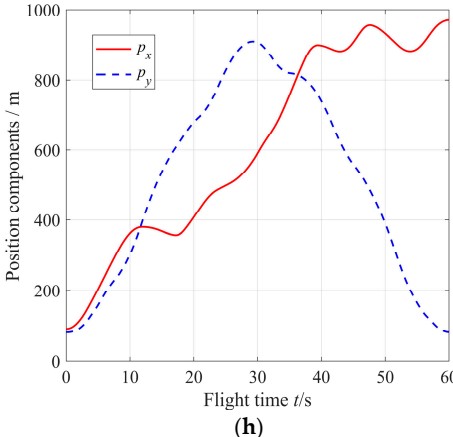

(**h**)

**Figure 10.** The results of Example 2. (**a**) Optimal trajectory solved by HNSCP, where the blue boxes indicate the full thrust arcs (12 arcs in total). (**b**) Optimal trajectory solved by HNSCP, where the red circles indicate the events when the vehicle appeared close to the obstacle boundary (14 events in total). (**c**) Trajectories obtained by HNSCP algorithm with different iteration numbers. (**d**) Monte Carlo random test trajectories. (**e**) Profiles of thrust acceleration magnitude, where the full thrust arcs have been marked with numbers. (**f**) Profiles of thrust direction. (**g**) Profiles of velocity components. (**h**) Profiles of position components.

We chose a small $\Delta\varepsilon$ to ensure that the algorithm had better convergence (100%). This adjustment significantly increased the number of iterations and, thus, the elapsed time. However, it can be seen from Table 8 that the algorithm could still converge within 2 s and had the potential for online applications. It was noted that in the single test, the TSCP algorithm was interrupted after the 24th iteration and failed to find a feasible solution.

From Figure 10a,e, it can be seen that the thrust acceleration profile calculated by HNSCP was the "bang-bang" form, and a total number of 12 maximum thrust arcs were found. Figure 10b shows the trajectory details, where the planned trajectory approached the constraint boundary 14 times (see from *A* to *N*), but none violated the constraint. The above results show that the algorithm had a strong adaptability and could still find a locally optimal feasible solution even in an extremely complex environment. Figure 10c shows the iterative details of the HNSCP algorithm. In the first few iterations, the trajectory traversed the obstacle. As the number of iterations increased, the trajectory gradually avoided all obstacles and converged to a feasible solution. Figure 10d shows the Monte Carlo test results. Starting from different positions, the HNSCP algorithm could find optimal feasible trajectories from the linear interpolation reference trajectory.

Due to the fitting error of the HNN, there may be the possibility of collision with real obstacles. Three strategies can be adopted to avoid such phenomena:

1. Set and increase the safety margin $d_{\text{mar}}$ (adopted in this work);
2. Increase the scale of the neural network to improve the fitting accuracy;
3. Take the HNSCP result as the initial value, and use the TSCP algorithm to refine the solution.

## 7. Conclusions

This work combines the sequential convex optimization algorithm with homotopy and neural network techniques. Compared with traditional methods, the proposed algorithm can deal with much more complex OACs and has significant advantages in convergence, real-time, and optimality. Numerical simulations show that for complex "maze" obstacle constraints, the proposed algorithm can achieve 100% convergence and find the local optimal solution within 2 s ($N = 100$). Consequently, it has the potential for onboard applications. Adding homotopy and neural network technologies has dramatically im-

proved convergence in practice, but lacks theoretical explanation. Future work will address this issue.

**Author Contributions:** Conceptualization, W.L. (Wenbo Li) and S.G.; methodology, W.L. (Wenbo Li) and W.L. (Wentao Li); software, W.L. (Wenbo Li) and W.L. (Wentao Li); validation, W.L. (Wenbo Li), S.G. and L.C.; formal analysis, W.L. (Wenbo Li); investigation, W.L. (Wenbo Li) and S.G.; resources, S.G. and L.C.; data curation, W.L. (Wenbo Li); writing—original draft preparation, W.L. (Wenbo Li) and W.L. (Wentao Li); writing—review and editing, S.G.; visualization, W.L. (Wenbo Li); supervision, S.G.; project administration, S.G.; funding acquisition, S.G. All authors have read and agreed to the published version of the manuscript.

**Funding:** This work was supported by the National Natural Science Foundation of China, grant numbers 11822205 and 11772167.

**Institutional Review Board Statement:** Not applicable.

**Informed Consent Statement:** Not applicable.

**Data Availability Statement:** Not applicable.

**Acknowledgments:** Not applicable.

**Conflicts of Interest:** The authors declare no conflict of interest.

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
