# Peer review of "Trajectory Optimization with Complex Obstacle Avoidance Constraints via Homotopy Network Sequential Convex Programming"

_aerospace, doi:10.3390/aerospace9110720_

Round 1

Reviewer 1 Report

In this paper, a trajectory optimization approach is presented to solve problems involving complex obstacle avoidance constraints by leveraging homotopy and neural networks. While the paper has some merit and is of interest to the general aerospace community, the technical details and results need some more clarity. Please address the following comments:

1. Trajectory optimization with obstacle avoidance is a pretty large field. It would be beneficial to add some more context in terms of literature. Some of the approaches that have been considered in the past which in some cases may be approached using convex optimization include

a.) using polynomial optimization to obtain obstacle models and using sequential convex optimization for trajectory optimization. For eg:

Misra, G., & Bai, X. (2020). Iteratively feasible optimal spacecraft guidance with non-convex path constraints using convex optimization. In AIAA Scitech 2020 Forum (p. 1350).

b.) mixed integer programming to solve obstacle avoidance trajectory planning problems. for eg:

Richards, Arthur, et al. "Spacecraft trajectory planning with avoidance constraints using mixed-integer linear programming." Journal of Guidance, Control, and Dynamics 25.4 (2002): 755-764.

2. In the introduction, it is very hard to gauge the motivation for using neural networks. Is it for dynamics approximation? for path constraint approximation? and Why are neural networks a good fit?

3. Eq. 7 seems a bit confusing. Is this a free final time problem? From the results it seemed that the horizon N was chosen, I am assuming t_f is known so is \Delta t a decision variable?

4. Eq. 15 and 16, are the obstacle avoidance constraints differentiable?

5. Can the first order Taylor approximation in Eq. 16 guarantee collision avoidance in between the discrete time-steps? Can this error be bounded?

6 . Why not compute the signed distance fields and its gradients offline and use them online? like in previous works such as STOMP trajectory optimization? What benefit does this neural network function provide?

7. Section 5.1, the description of Eikonal equation jumps out. It would be hard for the reader to understand the motivation behind using this approach. The authors should explain this section in more detail

8. The results look interesting and the authors demonstrate convergence via numerical examples. Would the inclusion of homotopy and neural networks impact the  theoretical convergence of the underlying SCP, which uses the penalty based trust region method? 

Reviewer 2 Report

 The authors proposed an effective algorithm that combines homotopy, network and convex programming to solve complicated trajectory optimization with obstacle avoidance problems. This manuscript is well written. This work is interesting for the community. I suggest to publish the work after minor revision. The following is the comments.

1. Lines 28-29, 'for example, in the rocket recovery guidance...' Any reference papers ?

2. Lines 42-43. Why the converged solution cannot guarantee feasibility? 

3. I think the 'homotopy state' is better to be changed to 'homotopy parameter'.

4. Line 186. What is the meaning of 'iso' in 'iso-\epsilon'?

5. Line 219-220. I'm confused by the virtual control varaible. The value of this variable should be 0 in the final solution because it actually does not exist in the dynamics. However, it seems that there is no such constraints.  

6. Line 222, Eq. (9). Please check the first equation. There should have v_{k+1}. Also, please write the second equation to a single line.

7. Line 245. 'inequations' should be 'inequality'

8. Line 249, eq. (16). Please check this equation. One \hat{p} seems to be lost.

9. Line 436. Do all the methods HNSCP, HOMO, CSTC and TSCP use the results from network ?

10. In table 5. I noticed that the parameter w_a = 100000, while w_u = 0.1. How to select the weights? Are the results largely affected by the weights ?

11. In Fig. 7a, the total iteration is 19 for HNSCP, while in table 6, the iteration is 17.

12. What's the value of homotopy parameter used in the first example ? In table 6, for HNSCP, the iterations are refer to the SCP iterations for each homotopy parameter? or total iterations of SCP? or total homotopy steps?

13. What's the training time for both examples ?

14. What's the computer langerage used to make the codes ?  

Round 2

Reviewer 1 Report

The authors have satisfactorily addressed my comments.